# GHPO: Adaptive Guidance for Stable and Efficient LLM Reinforcement Learning

## Abstract

Reinforcement Learning with Verifiable Rewards (RLVR) has recently emerged as a powerful paradigm for facilitating the self-improvement of large language models (LLMs), particularly in the domain of complex reasoning tasks. However, prevailing on-policy RL methods often contend with significant training instability and inefficiency. This is primarily due to a capacity-difficulty mismatch, where the complexity of training data frequently outpaces the model's current capabilities, leading to critically sparse reward signals and stalled learning progress. This challenge is particularly acute for smaller, more resource-efficient LLMs. To overcome this, we introduce the **G**uided **H**ybrid **P**olicy **O**ptimization (**GHPO**), a novel difficulty-aware reinforcement learning framework. GHPO dynamically calibrates task difficulty by employing adaptive prompt refinement to provide targeted guidance. This unique approach adaptively balances direct imitation learning for problems currently beyond the model's reach with exploration-based reinforcement learning for more manageable tasks, effectively creating a smooth and optimized learning curriculum. Extensive experiments demonstrate that GHPO achieves an average performance gain of approximately 5% across six challenging mathematics benchmarks, consistently outperforming strong on-policy reinforcement learning and curriculum learning baselines. Further analysis confirms that our framework significantly enhances both training stability and final reasoning performance, thus offering a scalable and efficient solution for developing powerful and robust reasoning models. The source code is available at: https://anonymous.4open.science/r/GHPO-ICLR-CD65/.

## 1 Introduction

A new generation of large reasoning models, including OpenAI-o3 OpenAI (2025), DeepSeek-R1 Guo et al. (2025), and Kimi-1.5 Team et al. (2025), is achieving state-of-the-art results in complex reasoning. A key characteristic of these models is their proficiency in producing extended Chains-of-Thought (CoT) Wei et al. (2022) and engaging in what appears to be reflective reasoning. This phenomenon, termed "test-time scaling", Guo et al. (2025) has proven highly effective for solving challenging mathematics and programming problems. At the heart of these achievements is a training methodology called ZERO-RL Guo et al. (2025). This paradigm employs Reinforcement Learning with Verifiable Rewards (RLVR)—a technique exemplified by DeepSeek-R1—to progressively enhance a base LLM's capabilities through reinforcement on its self-generated outputs.

Reinforcement learning (RL) based post-training has demonstrated superior generalization in enhancing reasoning capabilities compared to Supervised Fine-Tuning (SFT), which relies on imitation learning from high-quality, human-curated data or Chains of Thought (CoTs) distilled from more powerful models Chu et al. (2025). Nevertheless, current RL with Verifiable Rewards (RLVR) approaches, exemplified by Group Relative Policy Optimization (GRPO) Shao et al. (2024), confront substantial limitations. Being primarily on-policy, these methods exhibit a strong dependency on the current policy model's performance, leading to two major challenges: 1) *Reward Sparsity from Capacity-Difficulty Mismatch*: A significant mismatch between the inherent difficulty of the training data and the model's evolving capabilities often results in reward sparsity, where uniform and uninformative rewards impede the learning process Yu et al. (2025). 2) *Suboptimal Sample Efficiency*: On-policy algorithms frequently suffer from suboptimal sample efficiency. As training datasets incorporate increasingly challenging problems to drive benchmark performance, the policy

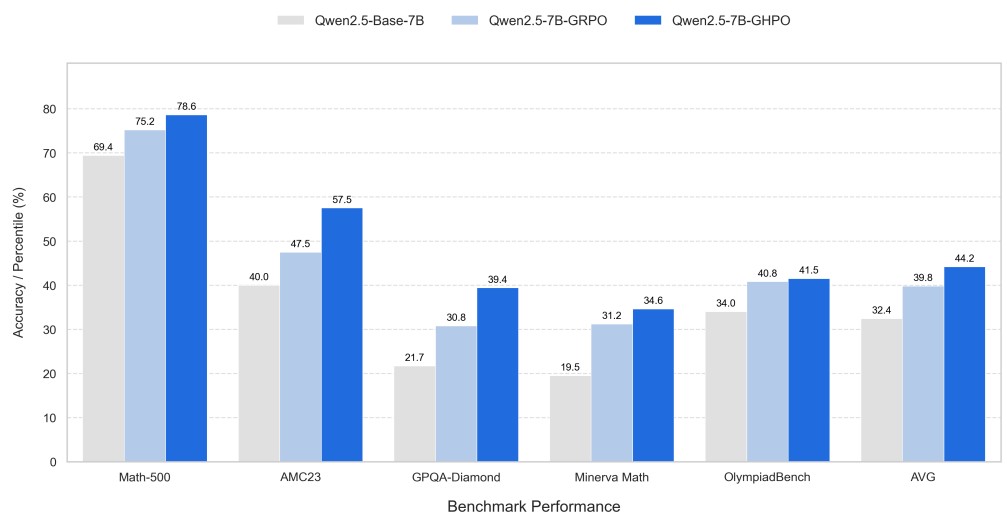

Figure 1: Overall Performance of GHPO across several benchmarks on the Qwen2.5-7B base model.

model struggles to learn effectively. This limitation is particularly pronounced for compact, on-device models with constrained capacities.

To address these limitations, several approaches have been proposed. Curriculum learning (CL) Luo et al. (2025), for instance, attempts to align task difficulty with the model's evolving capabilities by gradually introducing more complex samples. However, this often requires manual, heuristic-based partition of the dataset, which is not scalable. Other methods, like DAPO Yu et al. (2025), use dynamic sampling to filter out prompts that the model finds either too easy (perfect accuracy) or too hard (zero accuracy). Despite its effectiveness, this approach can be inefficient as it discards a significant portion of the training data. Another line of work explores off-policy or hybrid RL approaches to mitigate the instability inherent in on-policy learning Yan et al. (2025). These methods allow the policy model to learn from a broader distribution of responses but often require an auxiliary, resource-intensive LLM, thereby increasing training costs and complexity.

In this work, drawing inspiration from imitation learning techniques like SFT, we introduce a simple yet effective solution: guiding the model with partial ground truth solution traces. By conditioning the model on these traces, we steer its output distribution closer to the correct answer, which alleviates the reward sparsity problem for difficult samples. However, a naive application of this technique risks making the training data too easy, potentially reducing learning efficiency on problems the model could have solved independently. To address this, we propose the **G**uided **H**ybrid **P**olicy **O**ptimization (**GHPO**), which ingeniously combines online Reinforcement Learning (RL) and imitation learning within a unified framework. It uses a dynamic mechanism to first assess sample difficulty, then employs adaptive prompt refinement to provide varying levels of guidance. For problems the model can likely handle, GHPO primarily uses standard on-policy RL, encouraging exploration and self-discovery. But for more challenging samples, it seamlessly shifts to a form of imitation learning by offering explicit solution traces. This hybrid approach automatically balances exploration with direct guidance, preserving training efficiency for manageable tasks while effectively guiding the model through difficult ones, ultimately boosting both training stability and sample efficiency.

Our key contributions are:

- We identify the critical role of capacity alignment and reward sparsity problem in RLVR and propose the GHPO framework to improve training stability and efficiency.

- We introduce a novel framework to detect sample difficulty and adaptively switch between on-policy reinforcement learning and guided imitation learning by adaptive prompt refinement.

- We conduct extensive experiments on six mathematics benchmarks, demonstrating that GHPO outperforms state-of-the-art RL methods. Our results show consistent performance gains across different model families, validating the effectiveness and robustness of our approach.

## 2 PRELIMINARIES

### 2.1 REINFORCEMENT LEARNING WITH VERIFIABLE REWARDS (RLVR)

We begin by formally defining the problem of fine-tuning a Large Language Model (LLM) for complex reasoning tasks using RLVR. In this paradigm, an LLM, represented as a policy $\pi_\theta$ with parameters $\theta$, is trained to generate a sequence of tokens, or trajectory, $\tau = (o_1, o_2, \ldots, o_T)$ in response to an input prompt $q$. The process of generating this trajectory is modeled as a finite-horizon, token-level Markov Decision Process (MDP) Bellman & Kalaba (1957).

The components of this MDP are defined as follows:

- **State** ($s_t$): At each generation step $t$, the state is the concatenation of the initial prompt $q$ and the sequence of tokens generated thus far: $s_t = (q, o_1, o_2, \ldots, o_{t-1})$. The initial state is $s_0 = q$.

- **Action** ($a_t$): The action is the selection of the next token $o_t$ from the model's vocabulary $\mathcal{V}$.

- **Policy** ($\pi_\theta$): The policy is the LLM itself, which provides a probability distribution over the vocabulary for the next action (token) given the current state: $a_t = o_t \sim \pi_\theta(\cdot|s_t)$.

- **Reward** ($R$): RLVR employs sparse, terminal reward assigned by a verifier. A reward is assigned only at the end of a complete trajectory. A verifier determines if the final answer extracted from $\tau$ is correct, assigning a binary reward: $R = \begin{cases} 1 & \text{if the answer is correct} \\ 0 & \text{otherwise} \end{cases}$

The training objective is to learn the optimal policy parameters $\theta^*$ that maximize the expected terminal reward over the distribution of prompts $\mathcal{D}$, which is given by: $\mathcal{J}(\theta) = \mathbb{E}_{q \sim \mathcal{D}, \tau \sim \pi_\theta(\cdot|q)}[R(\tau)]$.

This objective $\mathcal{J}(\theta)$ is typically optimized using policy gradient algorithms, such as REINFORCE Sutton et al. (1999) or more advanced variants designed to handle the high variance and low sample efficiency inherent in this sparse-reward setting.

### 2.2 GROUP RELATIVE POLICY OPTIMIZATION (GRPO)

Group Relative Policy Optimization (GRPO) Shao et al. (2024) is a state-of-the-art RLVR algorithm that has demonstrated remarkable success in enhancing the reasoning abilities of LLMs, particularly for complex tasks like mathematics and programming Guo et al. (2025). Its core innovation is a novel method for advantage estimation that relies on intra-group comparison rather than absolute reward values. For a given prompt $q$, GRPO begins by sampling a group of $G$ distinct responses $\{o_i\}_{i=1}^{G}$ from policy model $\pi_{\theta_{\text{old}}}$. The advantage of each trajectory, $\hat{A}_{i,t}$, is then calculated by normalizing its reward $R_i$ against the statistics of the entire group: $\hat{A}_{i,t} = \frac{R_i - \mu_{\mathcal{R}}}{\sigma_{\mathcal{R}} + \epsilon}$, where $\mu_{\mathcal{R}} = \frac{1}{G} \sum_{j=1}^{G} R_j$ and $\sigma_{\mathcal{R}} = \sqrt{\frac{1}{G} \sum_{j=1}^{G} (R_j - \mu_{\mathcal{R}})^2}$ are the mean and standard deviation of rewards in the group, and $\epsilon$ is a small constant for numerical stability. A key aspect of this method is that this response-level advantage is shared across all token-generation steps within the same response $o_i$. The policy is then updated to increase the log-probability of tokens that belong to high-advantage trajectories.

### 2.3 THE CHALLENGE OF REWARD SPARSITY IN GRPO

Despite its successes, GRPO is susceptible to training inefficiencies and instability, a challenge often encountered in practical reproductions. We identify a primary cause for this fragility: a fundamental misalignment between the difficulty of training data and the capability of the policy model. This misalignment manifests as severe *reward sparsity*, which poses a significant obstacle to effective reinforcement learning.

This issue arises when a query $q$ is too difficult for the current policy $\pi_\theta$. In such cases, the model is likely to generate a group of $G$ responses where all trajectories are incorrect, yielding a reward vector of all zeros (i.e., $R_i = 0$ for all $i \in \{1, \ldots, G\}$). When all rewards in the group are zero, both the mean and standard deviation are also zero. Consequently, the advantage calculation from $\hat{A}_{i,t} = \frac{R_i - \mu_{\mathcal{R}}}{\sigma_{\mathcal{R}} + \epsilon}$ yields $\hat{A}_{i,t} = 0$ for all trajectories associated with that query.

The challenge of reward sparsity is particularly acute for capacity-constrained models, such as those designed for on-device deployment. To quantify this capacity-difficulty mismatch, we evaluated the performance of the Qwen2.5-7B-Instruct Qwen et al. (2025) model on the NuminaMath-1.5 LI et al. (2024) dataset, which comprises approximately 900,000 competition-level mathematics problems. Our analysis revealed that even the Qwen2.5-7B-Instruct model, a more capable version of its foundation model, failed to solve 52% of the problems. This significant finding indicates that a substantial portion of this dataset is far beyond the intrinsic reasoning capacity of the corresponding Qwen2.5-7B-Base model Qwen et al. (2025), let alone smaller LLMs with even more limited capabilities. This starkly illustrates the severity of the reward sparsity problem for on-device models. During reinforcement learning, over half of the dataset would be likely to yield zero-reward trajectories, providing no useful gradient signal and severely impeding the model's ability to learn.

## 3 THE PROPOSED FRAMEWORK

### 3.1 FROM STATIC GUIDANCE TO A DYNAMIC FRAMEWORK

As established, our core strategy is to integrate guidance directly into the reinforcement learning loop, conditioning the policy on partial ground-truth traces to overcome the reward sparsity detailed in Section 2.3. This approach is motivated by Assumption 1, which posits that such guidance increases the likelihood of success on difficult problems, thereby providing a valid learning signal where one would otherwise be absent. It's worth noting that ground truth guidance, in the form of solution traces, is often available for most mathematics data. However, during the RLVR training process, this valuable solution trace information is typically overlooked.

**Assumption 1** *Let $\mathcal{D}_{in}$ and $\mathcal{D}_{OOD}$ be the in-domain and out-of-distribution problem distributions, respectively. Let $\pi_{\theta_0}$ be a base policy. The OOD performance of any policy $\pi$ is measured by its expected reward $\mathcal{R}(\pi) := \mathbb{E}_{b \sim \mathcal{D}_{OOD}, \tau \sim \pi(\cdot|b)}[R(\tau)]$ on problem $b$.*

*Consider a problem $q \sim \mathcal{D}_{in}$ for which the base policy fails, i.e., its expected reward on this problem is non-positive ($\mathbb{E}_{\tau \sim \pi_{\theta_0}(\cdot|q)}[R(\tau)] \leq 0$). Let $h$ be a partial ground-truth solution trace for $q$.*

*Let two policies be fine-tuned from $\pi_{\theta_0}$ by maximizing an objective $\mathcal{J}_{GRPO}$:*

- *$\pi_{\theta_{q,h}}$, using the trace: $\theta_{q,h} = \arg\max_\theta \mathcal{J}_{GRPO}(\theta; \{(q, h)\})$*

- *$\pi_{\theta_q}$, without the trace: $\theta_q = \arg\max_\theta \mathcal{J}_{GRPO}(\theta; \{q\})$*

*We assume that using the ground-truth trace for a failing problem improves OOD generalization:*

$$\mathbb{E}_{b \sim \mathcal{D}_{OOD}, \tau \sim \pi_{\theta_{q,h}}(\cdot|b)}[R(\tau)] \geq \mathbb{E}_{b \sim \mathcal{D}_{OOD}, \tau \sim \pi_{\theta_q}(\cdot|b)}[R(\tau)]$$

And we demonstrate the effectiveness of this Assumption 1 through comprehensive experiment detailed in Section 4. By leveraging this property, we can obtain valid learning signals even on difficult problems that would otherwise yield zero rewards and vanishing gradients. However, a naive or static application of this principle—for instance, pre-labeling a fixed set of problems as "difficult" and always applying guidance to them—is suboptimal and suffers from two critical limitations:

- **Manual Curation and Scalability:** A static approach involves a laborious, offline process to determine when guidance is necessary for problems. This method is not only impractical at scale but also subjective and may not align perfectly with the model's actual knowledge gaps.

- **Evolving Model Capability:** A problem that is intractable for the policy at the beginning of training may become easy after several updates. A static guidance strategy cannot adapt to the model's evolving capabilities. It risks "over-guiding" the model on problems it could have solved through exploration, thereby stifling the learning of reasoning paths and reducing sample efficiency.

### 3.2 GUIDED HYBRID POLICY OPTIMIZATION (GHPO)

In this work, we propose **G**uided **H**ybrid **P**olicy **O**ptimization (**GHPO**), an automated framework designed to enhance reinforcement learning efficiency. As illustrated in Figure 2, GHPO dynamically

Figure 2: An illustration of the proposed GHPO framework.

assesses sample difficulty *on-the-fly* and adaptively switches between standard on-policy reinforcement learning and guided learning. This innovative approach ensures that guidance is provided only when truly necessary, preserving valuable exploration for problems within the model's current capabilities while providing adaptive refinement for more challenging scenarios.

GHPO is comprised of two core modules:

- **Automated Difficulty Detection**: This module assesses the inherent difficulty of the current problem, determining the subsequent learning process.

- **Adaptive Prompt Refinement**: Based on the detected difficulty, this module adaptively refines the prompt by incorporating different levels of ground truth guidance.

For a given query $q$ and ground truth answer $a$, GHPO first samples a group of $G$ individual responses, denoted as $\{o_i\}_{i=1}^{G}$. These responses are then evaluated by a reward model to obtain their corresponding binary rewards, $\{r_i\}_{i=1}^{G}$. Unlike GRPO, these group rewards are not directly used for advantage estimation. Instead, the difficulty detection module analyzes the sparsity of these group-level rewards. Based on this analysis, the corresponding prompt is refined according to a pre-specified strategy. Mathematically, GHPO optimizes the policy using the following objective:

$$\mathcal{J}_{\text{GHPO}}(\theta) = \mathbb{E}_{(q,a)\sim D,\ \{o_i\}_{i=1}^{G}\sim \pi_{\theta_{old}}(\cdot|q)}$$

$$\left[ \frac{1}{G}\sum_{i=1}^{G}\frac{1}{|o_i|}\sum_{t=1}^{|o_i|}\left(\min\left(r_{i,t}(\theta)\hat{A}_{i,t}, \text{clip}\left(r_{i,t}(\theta), 1-\epsilon, 1+\epsilon\right)\hat{A}_{i,t}\right) - \beta D_{\text{KL}}(\pi_\theta\|\pi_{\text{ref}})\right)\right]$$

(1)

$$\text{s.t.}\quad r_{i,t}(\theta) = \frac{\pi_\theta(o_{i,t}\mid q^*, o_{i,<t})}{\pi_{\theta_{\text{old}}}(o_{i,t}\mid q^*, o_{i,<t})},\qquad q^* = \begin{cases} q, & \text{if } \sum_{i=1}^{n} f(a, o_i) > 0 \\ q + \omega \cdot h_{f,q}, & \text{otherwise.} \end{cases}$$

(2)

where $f$ assesses whether the prediction is equivalent to the ground truth, $h_{f,q}$ is the full sequence of ground-truth solution for query $q$, and $\omega$ is the hint ratio adjusted by stages. By seamlessly integrating these two modules, our framework can efficiently switch between on-policy reinforcement learning and guided imitation learning, significantly enhancing training efficiency. We will detail these two modules further in the next section. An illustration of the prompt template in GHPO and a hint extraction example are provided in the Appendix B.1 and B.2, respectively.

### 3.3 AUTOMATED DIFFICULTY DETECTION

As highlighted in Assumption 1, incorporating offline ground-truth hints provides valuable learning signals even for the most challenging problems. However, a static, pre-defined strategy for identifying which problems need guidance is simply not scalable due to the limitations mentioned in Section 3.2. This method is not only impractical for large datasets but also inherently subjective, potentially missing the actual knowledge gaps of the model.

To address this, we propose a *difficulty detection module* that automatically identifies problem difficulty without manual intervention. Unlike other model-based approaches that demand high-cost computations from large language models for guidance, our method leverages the accuracy reward

inherent in the learning process: For each query $q$ in a batch, we assess its difficulty relative to the current policy model's capabilities ($\pi_\theta$) by analyzing the group rewards $\{r_i\}_{i=1}^G$ from $G$ individual responses for the same query, as formally defined in Equation (2). If all $G$ individual rewards are zero, the current policy model failed to generate a correct reasoning path despite $G$ sampling attempts from its output distribution. These sparse rewards yield no useful gradient for policy improvement. Such a query $q$ is thus identified as *difficult* for $\pi_\theta$, signaling the need for adaptive guidance.

### 3.4 ADAPTIVE PROMPT REFINEMENT WITH MULTI-STAGE GUIDANCE

The difficulty detection module identifies challenging queries where, as indicated by Assumption 1, incorporating ground-truth solution hints can provide valid learning signals. This guidance is applied by introducing a specific proportion of the ground-truth solution, quantified by the hint ratio parameter $\omega$. However, determining an optimal constant $\omega$ is challenging and often suboptimal, particularly for reasoning tasks with varying problem distributions, as more difficult problems inherently require a larger proportion of hints. To address this and ensure consistent learning for policy improvement, we propose an *Adaptive Prompt Refinement strategy with Multi-stage Guidance*, which dynamically adjusts the hint ratio $\omega$, with details provided in the Appendix B.3.

### 3.5 COLD-START STRATEGY

During the initial optimization stage, the policy model frequently struggles with adhering to specific formatting instructions, such as enclosing answers within a designated box. This often leads to a mismatch between predictions and ground-truth answers, resulting in low accuracy rewards. In such cases, the automated difficulty detection module might inaccurately classify the majority of queries as difficult, thereby introducing bias and unnecessarily consuming computational resources.

To address these issues, we propose an optional *cold-start strategy*. For the first $N$ optimization steps (specifically, we set $N = 20$ in our experiments), we temporarily disable the difficulty detection mechanism and instead apply the original GRPO training process. This approach not only conserves computational resources at the outset but also allows the model to develop fundamental formatting capabilities, preventing the introduction of early bias before adaptive guidance is implemented.

## 4 EXPERIMENT

### 4.1 TRAINING DETAILS

This study evaluates the proposed GHPO algorithm using verifiable mathematical tasks. While our method is designed for general applicability, its efficacy is demonstrated here within this domain. We constructed two training datasets of varying difficulty from the MATH Hendrycks et al. (2021b) and NuminaMath-1.5 LI et al. (2024) datasets: **Math3to5** and **NuminaMath-S**. To comprehensively evaluate GHPO, we selected two foundational large language models (LLMs), Qwen2.5-Base-7B Qwen et al. (2025) and Qwen2.5-Math-7B Yang et al. (2024), and implemented several RLVR methods, including GRPO, GRPO with curriculum learning (CL), and the proposed GHPO. Details on the dataset construction and the compared methods are provided in the Appendix C.1 and C.2.

We conducted experiments using the openr1 Face (2025) codebase and the TRL von Werra et al. (2020) framework. GHPO training was performed on 8 powerful GPUs, each equipped with 80GB of memory and high memory bandwidth. We evaluated performance on standard mathematical reasoning benchmarks, including: *MATH_500* Hendrycks et al. (2021a), *OlympiadBench* He et al. (2024), *Minerva Math* Lewkowycz et al. (2022), *GPQA-Diamond* Rein et al. (2023). Additionally, we assessed performance on competition-level benchmarks such as *AMC2023* and *AIME2024* Patel et al. (2024). Implementation and evaluation details are summarized in the Appendix C.3 and C.4.

### 4.2 OVERALL PERFORMANCE

Our experimental evaluation demonstrates the significant effectiveness and superiority of the proposed Guided Hybrid Policy Optimization (GHPO) method compared to conventional approaches. We present our findings across datasets of varying difficulty and with different baseline models.

Table 1: Performance comparison of models trained on the Math dataset.

| Model | AIME24 | Math-500 | OlympiadBench | AMC23 | Minerva Math | GPQA-Diamond | AVG |
|-------|--------|----------|---------------|-------|--------------|--------------|-----|
| Qwen2.5-Base-7B | 0.098 | 0.694 | 0.340 | 0.400 | 0.195 | 0.217 | 0.324 |
| Qwen2.5-Math-7B | 0.193 | 0.708 | 0.371 | 0.625 | 0.139 | 0.222 | 0.376 |
| Qwen2.5-7B-GRPO | 0.131 | 0.752 | 0.408 | 0.475 | 0.312 | 0.308 | 0.398 |
| Qwen2.5-7B-GHPO | 0.133 | 0.786 | 0.415 | 0.575 | 0.346 | 0.394 | 0.442 |

Table 2: Performance comparison of models trained on the Mixed dataset.

| Model | AIME24 | Math-500 | OlympiadBench | AMC23 | Minerva Math | GPQA-Diamond | AVG |
|-------|--------|----------|---------------|-------|--------------|--------------|-----|
| Qwen2.5-Base-7B | 0.098 | 0.694 | 0.340 | 0.400 | 0.195 | 0.217 | 0.324 |
| Qwen2.5-Math-7B | 0.193 | 0.708 | 0.371 | 0.625 | 0.139 | 0.222 | 0.376 |
| Qwen2.5-7B-GRPO | 0.122 | 0.774 | 0.396 | 0.525 | 0.283 | 0.353 | 0.409 |
| Qwen2.5-7B-GRPO-CL | 0.112 | 0.774 | 0.395 | 0.550 | 0.335 | 0.323 | 0.415 |
| Qwen2.5-7B-GRPO-CL-H(0.5) | 0.152 | 0.774 | 0.389 | 0.550 | 0.331 | 0.338 | 0.422 |
| Qwen2.5-7B-GHPO (Ours) | 0.163 | 0.776 | 0.389 | 0.575 | 0.342 | 0.404 | 0.442 |
| Qwen2.5-Math-7B-GRPO | 0.2698 | 0.81 | 0.4481 | 0.625 | 0.3456 | 0.3384 | 0.4728 |
| Qwen2.5-Math-7B-GHPO (Ours) | 0.3198 | 0.822 | 0.4525 | 0.7 | 0.3824 | 0.3687 | 0.5076 |

- **Performance on Medium-Difficulty Tasks**: We first conducted preliminary experiments to evaluate GHPO's performance on the `math3to5` training dataset. As detailed in Table 1, our proposed GHPO achieves an average accuracy improvement of **4.4%** over GRPO. This enhancement is primarily attributed to GHPO's ability to mitigate reward sparsity. The model trained using GHPO-based reinforcement learning consistently exhibits superior reasoning capabilities, achieving higher accuracy across all six evaluated benchmarks. Notably, for AMC2023 and GPQA-Diamond, GHPO yields more than **8%** improvement in accuracy compared to GRPO.

- **Performance on Challenging Tasks**: To further assess GHPO's robustness, we trained the Qwen2.5-7B base model using the more challenging NuminaMath-S dataset. In addition to the original GRPO, we introduced a curriculum learning baseline (Qwen2.5-7B-GRPO-CL), which manually partitions the training dataset by difficulty. As presented in Table 2, the Qwen2.5-7B-GHPO model consistently achieves superior performance over both Qwen2.5-7B-GRPO and Qwen2.5-7B-GRPO-CL. Specifically, GHPO demonstrates accuracy improvements across five of the six benchmarks when compared to both standard GRPO and GRPO with curriculum learning. This strongly indicates that reward sparsity significantly impedes effective learning, particularly for problems that lie beyond the model's current capabilities. For instance, in the highly challenging AIME2024 problems, where reward sparsity is pronounced due to the model's frequent inability to generate correct reasoning responses, our GHPO method achieves a substantial accuracy improvement ($0.122 \rightarrow 0.163$) over the original GRPO, despite using the identical training dataset.

- **Impact on curriculum learning with Fixed Hints**: While curriculum learning partially addresses the mismatch between model capability and problem difficulty, leading to an average accuracy improvement, it consistently falls short of our GHPO method. For a comprehensive comparison, we also investigated a scenario incorporating fixed hints. The Qwen2.5-7B-GHPO-CL-H0.5 model, which integrates a fixed 50% proportion of ground-truth solution traces into difficult problems combined with curriculum learning, shows an improvement over standard curriculum learning ($0.415 \rightarrow 0.422$) as shown in Table 2. However, its effectiveness remains lower than that of our proposed GHPO method (0.442).

## 4.3 GENERALIZATION TO OTHER MODELS

To further demonstrate the generalizability and robustness of our approach, we evaluated GHPO's effectiveness using a more capable base model: the Qwen2.5-Math-7B. This model, specifically designed and pre-trained for mathematical reasoning, offers a stronger foundation than the general-purpose Qwen2.5-Base-7B used in previous experiments.

As illustrated in Table 2, our proposed GHPO consistently produces a more powerful reasoning model, resulting in Qwen2.5-Math-7B-GHPO, when compared to the original GRPO method applied to the

same base model, Qwen2.5-Math-7B-GRPO. This sustained performance improvement with a more capable backbone affirms GHPO's benefits across different foundational model strengths and suggests its applicability beyond general-purpose LLMs, extending to specialized domains. This indicates that GHPO's adaptive guidance mechanism effectively complements even advanced pre-training, enabling more efficient and effective fine-tuning for complex reasoning tasks.

## 4.4 Training Dynamics

Understanding the behavior of our proposed GHPO framework during training is crucial to appreciating its advantages. Figure 3 illustrates the persistent challenge of reward sparsity: it shows the proportion of problems within a mini-batch that required the addition of hints to mitigate this issue. We observed that, during the initial training steps, a significant majority of problems proved too challenging for the current LLM to generate correct reasoning responses. This initial difficulty highlights the nascent capabilities of the model. Furthermore, this trend did not diminish rapidly; approximately 60% of problems continued to exceed the model's current capabilities even in subsequent training stages, underscoring the pervasive nature of the reward sparsity problem throughout the reinforcement learning (RL) process.

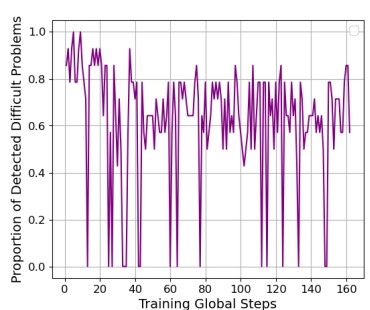

Figure 3: The proportion of problems detected to be difficult within a mini-batch.

To gain a deeper understanding of the distinct operational behaviors of GRPO and GHPO during training, we meticulously examined four representative metrics: format reward, accuracy reward, mean response length, and gradient norm. These dynamics are comprehensively depicted in Figure 4:

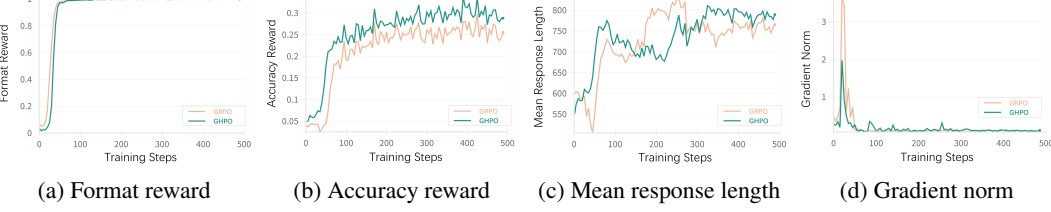

| (a) Format reward | (b) Accuracy reward | (c) Mean response length | (d) Gradient norm |

Figure 4: The metric curves of format reward, accuracy reward, mean response length, and gradient norm of GRPO and GHPO, which show the comparison of their RL training dynamics and serve as essential monitoring indicators.

- **Format Reward**: Both algorithms achieved comparable, near-maximal performance in terms of format reward early in training and sustained this high level throughout. This indicates that both methods are equally effective at enforcing desired structural constraints in the generated responses, such as adhering to specific answer formats.

- **Accuracy Reward**: Conversely, GHPO consistently exhibited a clear and substantial advantage in accuracy reward across all training stages. This direct superiority reflects GHPO's enhanced effectiveness in improving task-specific correctness, primarily attributable to its adaptive guidance mechanism which targets model knowledge gaps.

- **Mean Response Length**: Both methods demonstrated a steady increase in mean response length over time, which typically reflects the model's evolving ability to generate longer and potentially more sophisticated reasoning paths as training progresses. Notably, in later stages, GHPO generated significantly longer responses than GRPO. This observation suggests GHPO's enhanced capacity to construct more detailed and elaborate reasoning processes, possibly due to its exposure to partial ground-truth solutions that guide the expansion of logical steps.

- **Gradient Norm**: Finally, the gradient norm curves reveal that GHPO consistently maintained significantly smaller gradient magnitudes compared to GRPO. This is a critical indicator of a smoother and more stable optimization process for GHPO. Large gradient norms can signify

instability or oscillations in training, whereas GHPO's smaller norms suggest more controlled and efficient policy updates.

Collectively, these comprehensive observations demonstrate that GHPO not only yields superior task performance and promotes more detailed reasoning but also fosters a more stable and controlled optimization trajectory for policy updates throughout the training process.

## 5 RELATED WORK

Recent advances have demonstrated the remarkable success of reinforcement learning (RL) in enhancing the reasoning abilities of large language models (LLMs). A significant recent development is DeepSeek-R1 Guo et al. (2025), which introduced a pure reinforcement learning paradigm referred to as *zero RL training*. This approach utilizes a simple yet effective rule-based reward model for direct training from a base LLM. Building upon this foundation, subsequent efforts have progressively advanced this *zero RL training* methodology through reproduction, refinement, and further development. For instance, SimpleRL-Zoo Zeng et al. conducted a comprehensive empirical study to investigate zero RL training across diverse base models and sizes, aiming to elucidate its behavioral patterns and provide insights for future improvements.

Several variants have advanced *zero RL training* by refining core mechanisms or introducing novel designs to enhance performance and stability. DAPO Yu et al. (2025) analyzed the core mechanism behind zero RL training from the perspective of the policy optimization objective and introduced four key techniques to improve training efficiency, stability, and long CoT generation. Similarly, Dr. GRPO Liu et al. (2025) presented an unbiased optimization approach by removing length and standard deviation penalty terms from the original GRPO, enhancing token efficiency without sacrificing reasoning performance. As empirically demonstrated in Zeng et al., different base models exhibit distinct performance levels and behavioral patterns during RL training, since RL amplifies pre-existing sampling behaviors towards positive paths rather than fundamentally introducing novel capacities. To extend reasoning beyond intrinsic model limits, LUFFY Yan et al. (2025) balances imitation and exploration by augmenting on-policy zero RL training with off-policy reasoning demonstrations, incorporated as combined rollouts during training. In contrast to these value-model-free methods, VAPO Yue et al. (2025) introduced the first value-model-based RL training framework built upon PPO, integrating seven innovative techniques to enhance training stability and overall performance.

While these advancements have significantly propelled RL-based LLM training, a persistent challenge is the reward sparsity problem, especially in complex reasoning tasks where correct solutions are rare. To address this, various approaches have been proposed. Curriculum learning Luo et al. (2025), for instance, attempts to align task difficulty with the model's evolving capabilities by gradually introducing more complex samples. Other methods, such as DAPO Yu et al. (2025), utilize dynamic sampling to filter out prompts that the model finds either too easy or too hard. While effective in some contexts, this filtering approach can be inefficient, leading to the discarding of a significant portion of the valuable training data. Our proposed GHPO method directly confronts the reward sparsity issue by making full use of all training data through its adaptive difficulty detection and prompt refinement mechanisms, offering a more data-efficient and robust solution.

## 6 CONCLUSION

In this paper, we tackled the significant challenges of training instability and inefficiency in Reinforcement Learning with Verifiable Rewards (RLVR) for Large Language Models (LLMs). These issues primarily stem from a capacity-difficulty mismatch, which leads to sparse reward signals. To overcome this, we propose Guided Hybrid Policy Optimization (GHPO), a novel difficulty-aware RL framework. GHPO dynamically calibrates task difficulty through adaptive prompt refinement, balancing imitation learning for challenging problems with exploration-based RL for more manageable ones. Our extensive experiments show GHPO's significant superiority, achieving an average performance gain of approximately 5% across six challenging mathematics benchmarks. Furthermore, GHPO effectively enhances training stability. This framework offers a robust, scalable, and data-efficient solution for developing powerful reasoning LLMs by intelligently adapting the learning process to the model's evolving capabilities, leading to more stable and effective RL fine-tuning.

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

## A  STATEMENTS

### A.1  LLM USAGE STATEMENT

We used large language models (LLMs) solely for language editing and proofreading to improve the clarity and fluency of the manuscript. The models were not involved in research design, idea generation, data analysis, or technical content creation. All research contributions, including conceptualization, methodology, and writing, are entirely the work of the human authors.

### A.2  REPRODUCIBILITY STATEMENT

To ensure the reproducibility of our results, we have open-sourced both the implementation of our framework and the complete evaluation code. The code repository, including model training scripts, hyperparameter configurations, and data processing pipelines, is publicly available and linked in the abstract of this paper. All experimental settings are described in the main text and further detailed in the appendix.

### A.3  ETHICS STATEMENT

This work adheres to the ICLR Code of Ethics. The research does not involve human subjects, personal data, or sensitive content. All datasets used are publicly available and appropriately cited. There are no conflicts of interest, and all authors have contributed fairly to the work. We encourage transparency, reproducibility, and open scientific discussion.

## B  FURTHER DETAILS ON THE GHPO FRAMEWORK

### B.1  PROMPT TEMPLATE IN GHPO

An illustration of the proposed GHPO prompt template is shown in Fig. 5. Compared to the original GRPO prompt, GHPO introduces an enhanced structure where hints are extracted from the ground-truth solution and appended to the input problem with a guiding sentence: "**The following text is the beginning part of the answer, which you can refer to for solving the problem:**".

---

**GHPO Prompt**

```
<|im_start|>system
You are a helpful AI Assistant that provides well-reasoned and detailed responses.
You first think about the reasoning process as an internal monologue and then
provide the user with the answer. Respond in the following format:
<think>
...
<think>
<answer>
...
<answer><|im_end|>
<|im_start|>user
{problem}
The following text is the beginning part of the answer, which you can refer to for
solving the problem:
{hint}
<|im_end|>
<|im_start|>assistant
```

Figure 5: Prompt template used in the proposed GHPO framework.

## B.2 HINT EXTRACTION EXAMPLE

Figure 6 presents an illustration of how the proposed GHPO method effectively addresses a detected difficult problem. Consider the original question: "If a triangle has two sides of lengths 5 and 7 units, then how many different integer lengths can the third side be?" This particular problem proves challenging for the current model, which consistently fails to generate correct reasoning responses even after multiple sampling trajectories. This persistent failure leads directly to the problem of reward sparsity, where no valid gradient signals are available for learning.

To mitigate this, GHPO intelligently intervenes by extracting a partial ground-truth solution. In this example, 50% of the characters from the ground-truth solution are extracted as hints and appended to the original problem, along with a clear guiding sentence. As shown in the left portion of Figure 6, this process results in a newly constructed GHPO prompt that incorporates this targeted hint, thereby facilitating more effective and guided reasoning from the model.

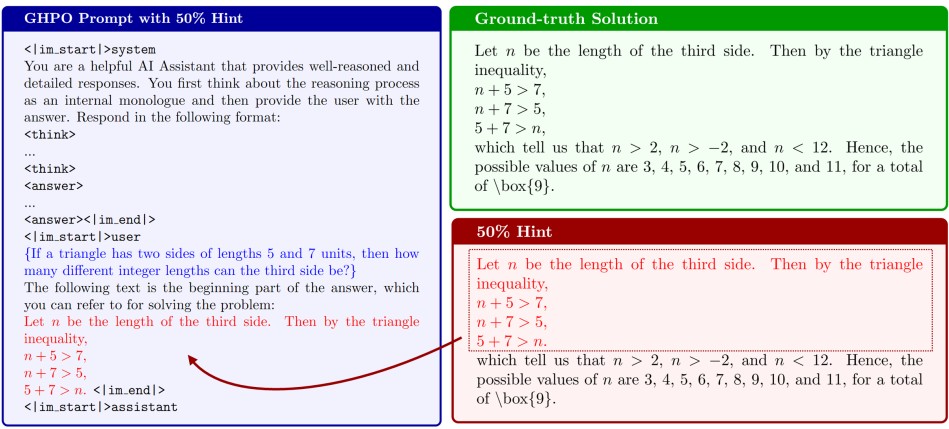

Figure 6: An illustration of using the proposed GHPO for addressing a detected difficult problem with 50% hint extraction.

## B.3 ADAPTIVE PROMPT REFINEMENT STRATEGY WITH MULTI-STAGE GUIDANCE

To address this and ensure consistent learning for policy improvement, we propose an *Adaptive Prompt Refinement strategy with Multi-stage Guidance*, which dynamically adjusts the hint ratio $\omega$. A key consideration is preventing redundant hints that oversimplify the task by excessively deviating from the original query. The core idea behind this dynamic hint injection is a linear schedule controlled by the learning stage. We begin by applying a small proportion of the ground-truth solution as an initial hint. If the model fails to generate a correct response, the hint's length is progressively increased in subsequent stages.

Specifically, during the first stage of the learning process, we apply a hint ratio of $\omega = 0.25$. This aims to balance data difficulty with the policy model's current capabilities. As defined in Equation (2), for queries identified as difficult within a training batch, 25% of the ground-truth solution traces $(h_{f,q})$ are extracted as assistance hints. These hints are then concatenated with the original query to form a refined query, $q^* = q + \omega \cdot h_{f,q}$, which is subsequently used for model inference. In the second stage, for queries where at least one generated response was evaluated as correct by the difficulty detection module, we retain the original query for subsequent group advantage computation. Conversely, if no correct response was generated, indicating a relatively harder query, we increase the hint ratio to $\omega = 0.5$ to provide further guidance. In this work, we utilize a maximum of three stages with a linear schedule for $\omega$: $\{0.25, 0.5, 0.75\}$. This strategy enables the full utilization of training data to enhance training efficiency, eliminating the need to directly remove difficult queries.

Beyond adaptively refining high-difficulty queries, this strategy can also be viewed as a dynamic data augmentation method. As the policy model's capabilities improve throughout the learning process, queries that initially required a high $\omega$ may eventually need only a lower level of guidance, or even no guidance at all.

# C EXPERIMENTAL SETTINGS

## C.1 TRAINING DATASETS

In the experiments, we constructed two training datasets of varying difficulty from the MATH Hendrycks et al. (2021b) and NuminaMath-1.5 LI et al. (2024) datasets:

- **Math3to5**: This dataset comprises 8,890 competition mathematics problems, ranging in difficulty from Level 3 to 5, each accompanied by a step-by-step ground-truth solution. This dataset represents a *medium difficulty level*.

- **NuminaMath-S**: A more challenging collection, this dataset contains 18,300 problems curated from NuminaMath-1.5. It integrates the math3to5 dataset with additional problems sourced from OlympiadBench and AMC. Each problem in this cleaned dataset also includes a step-by-step ground-truth solution. This dataset represents a *difficult level*.

These two datasets are designed to mirror the medium and difficult problem levels commonly encountered in real-world Reinforcement Learning with Verifiable Rewards (RLVR) training scenarios.

## C.2 COMPARED METHODS

To comprehensively evaluate GHPO, we selected two foundational large language models (LLMs) and implemented several RLVR methods:

- Qwen2.5-Base-7B Qwen et al. (2025): The foundational base model of Qwen2.5-7B.

- Qwen2.5-Math-7B Yang et al. (2024): A specialized mathematical LLM built upon Qwen2.5-7B.

- Qwen2.5-7B-GRPO: Qwen2.5-Base-7B fine-tuned using the GRPO training process Guo et al. (2025).

- Qwen2.5-Math-7B-GRPO: Qwen2.5-Math-7B fine-tuned using the GRPO training process.

- Qwen2.5-7B-GRPO-CL: This model undergoes a two-stage training process. Initially, Qwen2.5-Base-7B is trained with GRPO on the `math3to5` dataset. Subsequently, the resulting model is fine-tuned in a second stage using more challenging problems from OlympiadBench and AMC, significantly increasing the difficulty level from the first stage.

- Qwen2.5-7B-GRPO-CL-H: This variant applies a constant hint injection strategy during the second training stage of the Qwen2.5-7B-GRPO-CL pipeline, aiming to balance the learning process by providing consistent guidance. We apply constant hint inject to second training stage for balancing the learning process.

## C.3 IMPLEMENTATION DETAILS

We summarize the implementation details as follows:

- **Reward Setting**: For reward calculation, we employed a rule-based reward function that assigns a reward of +1 for correct answers and 0 for incorrect ones. Additionally, we incorporated a format-based reward to encourage the model to explicitly perform reasoning steps before delivering the final answer. This format-based reward assigns +1 if the response adheres to the "<think>...<think> <answer>...<answer>" format, and 0 otherwise. The weight ratio between the rule-based reward and the format-based reward was set to 2:1 for both the GRPO and GHPO methods.

- **Hyperparameters and Training Configuration**: We utilized the AdamW optimizer Loshchilov & Hutter (2017) with an initial learning rate of $1e \times 10^{-6}$. This learning rate was decayed to zero following a cosine schedule, including a warm-up phase of 10% of the total global steps. For training, the batch size was set to 112, with 8 responses sampled per query. We used 8 accumulation steps for gradient updates in each rollout. During the training stage, the sampling configurations included a temperature of 1.0 and a maximum generation of 2048 tokens per response. Notably, we did not use KL regularization losses or KL penalties in our rewards, as recent studies suggest their removal does not significantly impact performance Liu et al. (2025).

## C.4 EVALUATION DETAILS

We built our evaluation script using the Lighteval toolkit Habib et al. (2023) to assess the LLMs. During evaluation, we set the temperature to either 0.0 or 1.0 (depending on the benchmark's difficulty) and allowed a maximum generation length of 4096 tokens. To maintain consistency, we used the exact same prompt template as during training, with no extracted hints or hint-guided prompts utilized in this phase.

We evaluated performance on standard mathematical reasoning benchmarks, including: *MATH_500* Hendrycks et al. (2021a), *OlympiadBench* He et al. (2024), *Minerva Math* Lewkowycz et al. (2022), *GPQA-Diamond* Rein et al. (2023). Additionally, we assessed performance on competition-level benchmarks such as *AMC2023* and *AIME2024* Patel et al. (2024). For most benchmarks, we report *pass@1 results*. The pass@k accuracy represents the percentage of questions for which at least one correct response is obtained when generating $k$ responses per question. However, for AIME2024, which features fewer but more challenging problems, we report the average accuracy (avg@32), computed over 32 generated samples per problem.

# D CASE STUDY

We provide a detailed case study to illustrate the comparative effectiveness of GRPO and GHPO when addressing a particularly challenging mathematical problem, the specifics of which are presented in the Appendix. Table 3 displays this problem alongside its ground-truth solution.

When subjected to standard RL training under the original GRPO framework, the model frequently fails to generate correct reasoning paths, even after multiple sampling attempts. This consistent failure leads directly to a reward sparsity issue, which significantly impedes both the effectiveness and efficiency of the training process. Conversely, the proposed GHPO method addresses this challenge by intelligently employing a hint-guided prompt. As demonstrated by the example in Table 4, GHPO constructs a new problem input by concatenating the first 50% of the ground-truth solution into the original query. This targeted hint effectively guides the model, thereby enhancing its reasoning capabilities.

With this hint-guided prompt, the model successfully generates at least one correct reasoning path during GHPO-based RL training. The provided example of a correct response, obtained during GHPO training, clearly illustrates the model's adherence to the provided hint guidance, leading to a successful and accurate solution. In stark contrast, under standard GRPO training, which lacks any hint guidance, the model often generates reasoning paths that diverge significantly from the ground-truth solution. As exemplified by the incorrect answer in the Appendix, the absence of this crucial guidance leads to flawed reasoning and a failure to converge on the correct solution, underscoring the model's inherent limitations when

Table 3: The original problem and the ground-truth solution in the case study.

**Problem**:

Compute the sum

$$\sum_{i=0}^{\infty}\sum_{j=0}^{\infty}\frac{1}{(i+j+1)(i+j+2)(i+j+3)(i+j+4)(i+j+5)(i+j+6)(i+j+7)}.$$

**Ground-truth solution**:

First, we can write

$$\frac{1}{(i+j+1)(i+j+2)\cdots(i+j+6)(i+j+7)}=\frac{1}{6}\cdot\frac{(i+j+7)-(i+j+1)}{(i+j+1)(i+j+2)\cdots(i+j+6)(i+j+7)}$$

$$=\frac{1}{6}\left(\frac{1}{(i+j+1)(i+j+2)\cdots(i+j+6)}-\frac{1}{(i+j+2)\cdots(i+j+6)(i+j+7)}\right).$$

Thus, the following sum telescopes:

$$\sum_{j=0}^{\infty}\frac{1}{(i+j+1)(i+j+2)\cdots(i+j+6)(i+j+7)}$$

$$=\sum_{j=0}^{\infty}\frac{1}{6}\left(\frac{1}{(i+j+1)(i+j+2)\cdots(i+j+6)}-\frac{1}{(i+j+2)\cdots(i+j+6)(i+j+7)}\right)$$

$$=\frac{1}{6}\left(\frac{1}{(i+1)\cdots(i+6)}-\frac{1}{(i+2)\cdots(i+7)}\right)+\frac{1}{6}\left(\frac{1}{(i+2)\cdots(i+7)}-\frac{1}{(i+3)\cdots(i+8)}\right)$$

$$+\frac{1}{6}\left(\frac{1}{(i+3)\cdots(i+8)}-\frac{1}{(i+4)\cdots(i+9)}\right)+\cdots=\frac{1}{6(i+1)(i+2)\cdots(i+5)(i+6)}.$$

We can then write

$$\frac{1}{6(i+1)(i+2)\cdots(i+5)(i+6)}=\frac{1}{5}\cdot\frac{(i+6)-(i+1)}{6(i+1)(i+2)\cdots(i+5)(i+6)}$$

$$=\frac{1}{30}\left(\frac{1}{(i+1)(i+2)(i+3)(i+4)(i+5)}-\frac{1}{(i+2)(i+3)(i+4)(i+5)(i+6)}\right).$$

We obtain another telescoping sum:

$$\sum_{i=0}^{\infty}\frac{1}{6(i+1)(i+2)\cdots(i+5)(i+6)}$$

$$=\sum_{i=0}^{\infty}\frac{1}{30}\left(\frac{1}{(i+1)(i+2)(i+3)(i+4)(i+5)}-\frac{1}{(i+2)(i+3)(i+4)(i+5)(i+6)}\right)$$

$$=\frac{1}{30}\left(\frac{1}{(1)(2)(3)(4)(5)}-\frac{1}{(2)(3)(4)(5)(6)}\right)+\frac{1}{30}\left(\frac{1}{(2)(3)(4)(5)(6)}-\frac{1}{(3)(4)(5)(6)(7)}\right)$$

$$+\frac{1}{30}\left(\frac{1}{(3)(4)(5)(6)(7)}-\frac{1}{(4)(5)(6)(7)(8)}\right)+\cdots=\frac{1}{30}\cdot\frac{1}{(1)(2)(3)(4)(5)}=\boxed{\frac{1}{3600}}.$$

Table 4: The extracted 50% hint and the constructed new problem for GHPO.

**Extracted 50% hint**:

First, we can write

$$\frac{1}{(i+j+1)(i+j+2)\cdots(i+j+6)(i+j+7)} = \frac{1}{6} \cdot \frac{(i+j+7)-(i+j+1)}{(i+j+1)(i+j+2)\cdots(i+j+6)(i+j+7)}$$

$$= \frac{1}{6}\left( \frac{1}{(i+j+1)(i+j+2)\cdots(i+j+6)} - \frac{1}{(i+j+2)\cdots(i+j+6)(i+j+7)} \right).$$

Thus, the following sum telescopes:

$$\sum_{j=0}^{\infty} \frac{1}{(i+j+1)(i+j+2)\cdots(i+j+6)(i+j+7)}$$

$$= \sum_{j=0}^{\infty} \frac{1}{6}\left( \frac{1}{(i+j+1)(i+j+2)\cdots(i+j+6)} - \frac{1}{(i+j+2)\cdots(i+j+6)(i+j+7)} \right)$$

$$= \frac{1}{6}\left( \frac{1}{(i+1)\cdots(i+6)} - \frac{1}{(i+2)\cdots(i+7)} \right) + \frac{1}{6}\left( \frac{1}{(i+2)\cdots(i+7)} - \frac{1}{(i+3)\cdots(i+8)} \right)$$

$$+ \frac{1}{6}\left( \frac{1}{(i+3)\cdots(i+8)} - \frac{1}{(i+4)\cdots(i+9)} \right) + \cdots$$

**New Problem with 50% hint**:

Compute the sum

$$\sum_{i=0}^{\infty}\sum_{j=0}^{\infty} \frac{1}{(i+j+1)(i+j+2)(i+j+3)(i+j+4)(i+j+5)(i+j+6)(i+j+7)}.$$

**The following text is the beginning part of the answer, which you can refer to for solving the problem:**

First, we can write

$$\frac{1}{(i+j+1)(i+j+2)\cdots(i+j+6)(i+j+7)} = \frac{1}{6} \cdot \frac{(i+j+7)-(i+j+1)}{(i+j+1)(i+j+2)\cdots(i+j+6)(i+j+7)}$$

$$= \frac{1}{6}\left( \frac{1}{(i+j+1)(i+j+2)\cdots(i+j+6)} - \frac{1}{(i+j+2)\cdots(i+j+6)(i+j+7)} \right).$$

Thus, the following sum telescopes:

$$\sum_{j=0}^{\infty} \frac{1}{(i+j+1)(i+j+2)\cdots(i+j+6)(i+j+7)}$$

$$= \sum_{j=0}^{\infty} \frac{1}{6}\left( \frac{1}{(i+j+1)(i+j+2)\cdots(i+j+6)} - \frac{1}{(i+j+2)\cdots(i+j+6)(i+j+7)} \right)$$

$$= \frac{1}{6}\left( \frac{1}{(i+1)\cdots(i+6)} - \frac{1}{(i+2)\cdots(i+7)} \right) + \frac{1}{6}\left( \frac{1}{(i+2)\cdots(i+7)} - \frac{1}{(i+3)\cdots(i+8)} \right)$$

$$+ \frac{1}{6}\left( \frac{1}{(i+3)\cdots(i+8)} - \frac{1}{(i+4)\cdots(i+9)} \right) + \cdots$$

**Correct Answer (GHPO)**

To solve the given double sum, we start by simplifying the inner sum. The expression inside the inner sum is a product of several consecutive integers in the denominator, which suggests a telescoping series. We can decompose the fraction in the inner sum to facilitate this.

We rewrite the inner sum as follows:

$$\frac{1}{(i+j+1)(i+j+2)\cdots(i+j+7)}.$$

We can express this fraction using partial fractions. Specifically, we can write:

$$\frac{1}{(i+j+1)(i+j+2)\cdots(i+j+7)}$$
$$= \frac{1}{6}\left(\frac{1}{(i+j+1)(i+j+2)\cdots(i+j+6)} - \frac{1}{(i+j+2)\cdots(i+j+6)(i+j+7)}\right).$$

This decomposition allows us to see that the inner sum will telescope.
Substituting this decomposition into the inner sum, we get:

$$\sum_{j=0}^{\infty}\frac{1}{(i+j+1)(i+j+2)\cdots(i+j+6)(i+j+7)}$$
$$= \sum_{j=0}^{\infty}\frac{1}{6}\left(\frac{1}{(i+j+1)(i+j+2)\cdots(i+j+6)} - \frac{1}{(i+j+2)\cdots(i+j+6)(i+j+7)}\right).$$

This sum telescopes, and most terms cancel out, leaving us with:

$$\frac{1}{6}\left(\frac{1}{(i+1)\cdots(i+6)} - \frac{1}{(i+2)\cdots(i+7)} + \frac{1}{(i+2)\cdots(i+7)} - \frac{1}{(i+3)\cdots(i+8)} + \cdots\right).$$

The resulting sum is:

$$\frac{1}{6(i+1)(i+2)\cdots(i+5)(i+6)}.$$

We now need to evaluate the outer sum:

$$\sum_{i=0}^{\infty}\frac{1}{6(i+1)(i+2)\cdots(i+5)(i+6)}.$$

We decompose the fraction as follows:

$$\frac{1}{6(i+1)(i+2)\cdots(i+5)(i+6)} = \frac{1}{30}\left(\frac{1}{(i+1)(i+2)\cdots(i+5)} - \frac{1}{(i+2)(i+3)\cdots(i+6)}\right).$$

This decomposition also results in a telescoping sum.
Substituting this decomposition into the outer sum, we get:

$$\sum_{i=0}^{\infty}\frac{1}{6(i+1)(i+2)\cdots(i+5)(i+6)} = \sum_{i=0}^{\infty}\frac{1}{30}\left(\frac{1}{(i+1)(i+2)\cdots(i+5)} - \frac{1}{(i+2)(i+3)\cdots(i+6)}\right).$$

This sum telescopes, and most terms cancel out, leaving us with:

$$\frac{1}{30}\left(\frac{1}{1\cdot2\cdot3\cdot4\cdot5}\right) = \frac{1}{30\cdot120} = \boxed{\frac{1}{3600}}.$$

**Incorrect Answer (GRPO)**

To compute the sum

$$\sum_{i=0}^{\infty}\sum_{j=0}^{\infty}\frac{1}{(i+j+1)(i+j+2)(i+j+3)(i+j+4)(i+j+5)(i+j+6)(i+j+7)},$$

we can simplify the problem by introducing a new variable $k = i + j$. This transforms the double sum into a single sum over $k$.

The sum can be rewritten as:

$$\sum_{k=0}^{\infty}\sum_{i+j=k}\frac{1}{(k+1)(k+2)(k+3)(k+4)(k+5)(k+6)(k+7)}.$$

For each fixed $k$, the inner sum $\sum_{i+j=k} 1$ simply counts the number of pairs $(i,j)$ such that $i + j = k$. Since $i$ and $j$ are non-negative integers, there are $k + 1$ such pairs (namely $(0, k), (1, k-1), \ldots, (k, 0)$).

So the sum becomes:

$$\sum_{k=0}^{\infty}\cdot\frac{k+1}{(k+1)(k+2)(k+3)(k+4)(k+5)(k+6)(k+7)} = \sum_{k=0}^{\infty}\frac{1}{(k+2)(k+3)(k+4)(k+5)(k+6)(k+7)}.$$

To evaluate the sum

$$\sum_{k=0}^{\infty}\frac{1}{(k+2)(k+3)(k+4)(k+5)(k+6)(k+7)},$$

we can use partial fraction decomposition. Notice that:

$$\frac{1}{(k+2)(k+3)(k+4)(k+5)(k+6)(k+7)} = \frac{A}{k+2} + \frac{B}{k+3} + \frac{C}{k+4} + \frac{D}{k+5} + \frac{E}{k+6} + \frac{F}{k+7}.$$

By solving for the coefficients $A, B, C, D, E, F$, we can find the partial fraction decomposition. However, a simpler approach is to recognize that this is a telescoping series.

The series

$$\sum_{k=0}^{\infty}\frac{1}{(k+2)(k+3)(k+4)(k+5)(k+6)(k+7)}$$

can be evaluated by recognizing that it is a telescoping series. The general term

$$\frac{1}{(k+2)(k+3)(k+4)(k+5)(k+6)(k+7)}$$

can be written in a form that allows for telescoping.

To see this, consider the partial sum:

$$S_n = \sum_{k=0}^{n}\frac{1}{(k+2)(k+3)(k+4)(k+5)(k+6)(k+7)}.$$

The series telescopes, and the sum simplifies to:

$$S_n = \frac{1}{5\cdot 6\cdot 7} - \frac{1}{(n+3)(n+4)(n+5)(n+6)(n+7)}.$$

As $n \to \infty$, the second term goes to 0, so the sum of the infinite series is:

$$\sum_{k=0}^{\infty}\frac{1}{(k+2)(k+3)(k+4)(k+5)(k+6)(k+7)} = \frac{1}{5\cdot 6\cdot 7} = \frac{1}{210}.$$

The final answer is:

$$\boxed{\frac{1}{210}}$$

