# OpenReview forum: "GHPO: Adaptive Guidance for Stable and Efficient LLM Reinforcement Learning"
_ICLR.cc/2026/Conference — ICLR 2026 Conference Withdrawn Submission_

### Official Review · Reviewer_EM4V · 2025-10-25

**Soundness:** 3
**Presentation:** 3
**Contribution:** 3
**Rating:** 6
**Confidence:** 3

**Summary:**

This paper proposes GHPO, an adaptive guidance method designed to improve the stability and efficiency of LLM reinforcement learning. The authors observe that in RLVR, the model often wastes training instances that are too difficult to solve. GHPO addresses this by adaptively providing hints for such challenging instances, enabling a weaker model to leverage these hints to solve the problems rather than discarding them. The experimental results are promising.

**Strengths:**

1. The idea of using adaptive hints to improve RL stability and efficiency is interesting and intuitive.
2. The proposed GHPO achives better performance than the GRPO/GRPO-CL baselines.

**Weaknesses:**

1. The paper criticizes DAPO for discarding a significant portion of training data. But DAPO can potentially reuse those discarded training data at later stages when the model improves. So the criticism may not be entirely fair.
2. The experiments do not include an ablation study for DAPO, which limits the completeness of the comparison.
3. The proposed method may not generalize to datasets where the ground-truth solution lacks a chain-of-thought (CoT) component. For example, in multiple-choice problems with answers like (A, B, C, D), the hint cannot be extracted from the final solution.

**Questions:**

When comparing to GRPO, the x-axis represents training steps. Do GHPO and GRPO use the same amount of compute per training step? If not, could the authors include a comparison where the x-axis measures compute or wall-clock time instead?

---

### Official Review · Reviewer_PcPV · 2025-10-30

**Soundness:** 1
**Presentation:** 3
**Contribution:** 1
**Rating:** 2
**Confidence:** 3

**Summary:**

This paper presents Guided Hybrid Policy Optimization (GHPO), a framework designed to improve the stability and efficiency of Reinforcement Learning with Verifiable Rewards (RLVR) for LLM reasoning tasks . The authors posit that a "capacity-difficulty mismatch" leads to sparse rewards, which stalls learning. GHPO addresses this by first detecting "difficult" samples—those for which the policy model generates only incorrect responses (all-zero rewards) . For these samples, the framework employs "adaptive prompt refinement," which injects a partial ground-truth solution trace (a "hint") into the prompt, effectively switching from exploration-based RL to a guided imitation learning .

**Strengths:**

- The paper clearly identifies a critical and practical failure mode in on-policy RLVR (Section 2.3). It correctly points out that when all $G$ responses in a group receive a zero reward, both the mean and standard deviation of the rewards become zero, which nullifies the advantage signal and provides no gradient for the update. GHPO's mechanism of using this "all-zero" state as an online difficulty detector is pragmatic and intuitive.
- The paper is well-written, and the method is explained clearly, with excellent support in the appendices. The authors provide the exact prompt template, a concrete example of hint extraction (Appendix B.2, Figure 6), and a detailed case study (Appendix D).
- The paper provides direct evidence for its stability claims in Section 4.4. Figure 4(d) clearly illustrates that GHPO maintains a significantly smaller and more stable gradient norm throughout training compared to GRPO, which suffers from larger spikes.

**Weaknesses:**

- Weak Baseline Comparisons: The paper's empirical evaluation is limited by comparing GHPO almost exclusively to variants of GRPO (standard GRPO and GRPO-CL). Without comparisons to these state-of-the-art methods (LUFFY, UFT, ...), it is impossible to situate GHPO's performance and determine if its gains are a significant advancement over the field or merely an improvement on a specific, and possibly simple, baseline.
- The core idea of GHPO is to "adaptively balance direct imitation learning... with exploration-based reinforcement learning". This concept of a hybrid SFT/RL framework is not new in UFT. This sounds conceptually very similar to GHPO's approach of falling back on SFT (imitation) when RL (exploration) fails. The paper fails to discuss the conceptual differences between GHPO and UFT and, as noted, does not include it as an experimental baseline, making the novelty of the proposed framework unclear.
-  The paper frames GHPO as a "novel difficulty-aware reinforcement learning framework". However, the described mechanism does not appear to modify the core policy optimization algorithm itself. Rather, it acts as a data-side, heuristic-based pre-processing loop: if the RL algorithm fails to produce any positive reward for a sample $q$, then the sample is modified to $q*$ by injecting a hint, and the same RL algorithm is then applied to $q*$. This is more accurately described as an online data augmentation or dynamic curriculum strategy. This framing creates a methodological tension, as the paper critiques SFT (imitation learning) while its own "novel" mechanism fundamentally relies on reverting to SFT to provide the learning signal for difficult problems.
- By providing partial ground-truth solutions for all failed problems, it is unclear what skill the model is acquiring. The paper claims to teach "reasoning", but the mechanism may instead be teaching a more brittle pattern-matching skill of "how to complete a prompt that already contains the first 25-75% of the correct answer". This is a form of hint-based imitation, and the paper provides no evidence that the model learns to generalize this reasoning without the hints. The case study in Appendix D seems to confirm this, as the GHPO-generated correct answer is a direct, logical continuation of the 50% hint provided.
- The paper claims efficiency, but the described algorithm (Figure 2) implies a significant computational overhead. For difficult samples, which Figure 3 suggests is the majority of the data (~60%), the framework must perform two sets of forward passes: one (with $G$ samples) on the original $q$ to detect failure, and a second (with $G$ samples) on the modified $q^*$ to generate trajectories for the update. This overhead is never quantified or discussed. Additionally, the paper lacks a critical ablation study for its core components, such as automated difficulty detection.

[1] UFT: Unifying Supervised and Reinforcement Fine-Tuning

**Questions:**

See weaknesses.

---

### Official Review · Reviewer_BfxR · 2025-10-31

**Soundness:** 3
**Presentation:** 2
**Contribution:** 3
**Rating:** 2
**Confidence:** 4

**Summary:**

To address the instability and inefficiency issues observed in existing reinforcement learning methods with verifiable rewards (such as GRPO), this paper proposes a novel difficulty-aware reinforcement learning framework, GHPO. Experimental results demonstrate that GHPO achieves significant performance improvements across multiple benchmark tests, confirming its effectiveness in enhancing the reasoning capabilities of large language models (LLMs). Overall, the paper presents an innovative idea with solid empirical results, but it still requires further refinement in terms of writing clarity and citation consistency to improve its overall academic quality.

**Strengths:**

**Novel approach:**
The method combines imitation learning and reinforcement learning in a dynamic way and introduces a multi-stage hint mechanism to improve training stability and efficiency

**Comprehensive experimentation:**
The authors evaluate the approach on two types of training datasets (medium and hard), two base models (Qwen2.5-Base-7B and Qwen2.5-Math-7B), and several benchmarks (Math_500, OlympiadBench, AIME). GHPO shows consistent improvements over GRPO and GRPO+CL.

**Weaknesses:**

1. The citation and reference formatting do not follow ICLR standards. Several entries lack venue, volume, or page information, and some published works (e.g., DeepSeek-R1, OlympiadBench) are not properly cited, which affects the paper’s professionalism.
2. The paper does not provide any sensitivity or ablation study for key hyperparameters such as the hint ratio (ω) or the number of training stages, making it unclear how robust the method is to these design choices.
3. There is insufficient discussion of computational cost — the additional overhead from adaptive difficulty detection and multi-stage hint generation is not quantified.
4. There are minor writing and formatting issues, including occasional missing punctuation (e.g., missing period at the end of the second paragraph on page 3), which reduce readability.

**Questions:**

1. The method introduces new hyperparameters, such as the scheduling strategy for the hint ratio ω and stage transition thresholds. What is the sensitivity of these hyperparameters across different models or datasets?

2. How does GHPO handle "absolutely difficult" samples where the model consistently fails even with the maximum hint ratio (ω=0.75) during training?

---

### Official Review · Reviewer_5gfq · 2025-10-31

**Soundness:** 3
**Presentation:** 3
**Contribution:** 3
**Rating:** 6
**Confidence:** 3

**Summary:**

The paper proposes GHPO (Guided Hybrid Policy Optimization), a difficulty-aware RLVR framework that detects hard samples on the fly and injects partial solution traces via adaptive prompt refinement. GHPO switches between standard on-policy RL for manageable items and imitation-like guided training for difficult ones, scheduling the hint ratio across stages to form a smooth learning curriculum. On six math benchmarks, the authors report +5% average gains over strong on-policy RL and curriculum baselines, along with improved training stability.

**Strengths:**

1. Directly tackles reward sparsity with a simple mechanism (difficulty detection + adaptive hints) that keeps training on-policy and lightweight.
2. Empirical gains and stability. The paper reports consistent improvements (+5% across six math benchmarks) and shows training-dynamics curves where GHPO achieves higher accuracy reward, longer reasoning traces, and smaller gradient norms than GRPO.

**Weaknesses:**

1. Brittle difficulty detector. The “all-zero” rule (difficult iff all G rewards are 0) is too hard; with G=8, a single lucky success disables guidance. ~60% of problems are still flagged as difficult during training, suggesting heavy hint reliance under a rigid threshold.
2. Unvalidated cold start. Detection is disabled for the first 20 steps (pure GRPO), which is reasonable but ad-hoc; there is no ablation of N.

**Questions:**

1. Detector design. Will you compare the all-zero rule with soft thresholds (e.g., success-rate < 1/8 or 2/8)? Please report impacts on hint usage, accuracy, and stability.
2. Stability of GPHO. Will you provide mean ± std over multiple independent runs (matched compute) for GHPO and baselines to substantiate stability claims?
3. Cold-start ablation. Will you ablate the cold-start window and show effects on final accuracy and the fraction-difficult curve?

---

### Note · Authors · 2025-11-26

I have read and agree with the venue's withdrawal policy on behalf of myself and my co-authors.